# Park entrances, commonly contaminated with infective *Toxocara canis* eggs, present a risk of zoonotic infection and an opportunity for focused intervention

**Jason D. Keegan**[1]*, **Paul M. Airs**[2], **Claire Brown**[2], **Anya Rishi Dingley**[1], **Conor Courtney**[1], **Eric R. Morgan**[2], **Celia V. Holland**[1]

**1** Parasitology Research Group, Department of Zoology, Trinity College, Dublin, Ireland, **2** School of Biological Sciences, Queen's University Belfast, Belfast, United Kingdom

* devoykej@tcd.ie

## Abstract

*Toxocara canis* and *Toxocara cati*, the common roundworms of dogs and cats respectively are commonly found in the soil of public parks. This zoonotic parasite can also infect humans causing difficult to diagnose diseases. Direct contact with contaminated soil is considered one of the primary routes of transmission from animals to humans with the contamination of public places common around the world. In this study we aimed to employ an easily repeatable soil sampling methodology to identify differences in contamination levels between and within parks. Egg density was found to differ significantly between parks and park entrances were the most heavily contaminated locations within parks, followed by playgrounds. Species confirmation by polymerase chain reaction was conducted on a proportion of the recovered eggs and identified most as *T. canis* (n=36) while three eggs were identified as T. *cati*. These results indicate that dogs are responsible for the majority of environmental contamination in Dublin City parks, with the highest levels of contamination found around park entrances. Concentration of preventive efforts on dog fouling at these sites is recommended to reduce risks of zoonotic disease transmission.

## Author summary

The common roundworms of dogs and cats are also capable of infecting other animals, including humans. Infected dogs and cats can release parasite eggs into the environment, where they develop if the conditions are warm and moist enough. Accidental ingestion of these eggs can cause difficult to diagnose illness in humans. The aim of this study was to measure the number of *Toxocara* spp. eggs in soil samples of public parks in Dublin City. We aimed to do this in a clear and repeatable manner, using freely available software to help standardise the selection of soil sampling locations, as well as providing a detailed methodology regarding how soil samples were actually taken, information which is often lacking from similar studies. In addition, we aimed to establish the species of each

**Data availability statement:** All relevant data are within the manuscript and its Supporting information files.

**Funding:** This research was funded by The Irish Research Council's Postdoctoral Fellowship programme (GOIPD/2020/510 to JDK and CVH). https://research.ie/ The funders had no role in study design, data collection and analysis, decision to publish, or preparation of the manuscript.

**Competing interests:** The authors have declared that no competing interests exist.

egg detected using PCR, to identify whether dogs or cats were likely responsible for the presence of eggs in the environment. Based on the results of this study, park entrances were the most commonly contaminated location within each park. The majority of eggs detected were potentially infective, and dogs were responsible for the majority of this contamination.

## Introduction

Dogs and cats, our closest animal companions, are commonly infected with the roundworm parasites, *Toxocara canis* and *Toxocara cati*. Recent meta-analyses have shown that 11.1% of the world's dogs [1] and 17% of cats [2] are infected with their respective roundworms *T. canis* and *T. cati*. Adult *Toxocara* worms in the intestines of their definitive hosts release large numbers of eggs into the environment with the animal's faeces. After some time at the appropriate temperature and humidity levels, these eggs can become infective to dogs, cats and other hosts, including humans. Contamination of the environment is also commonplace, again emphasised by a recent meta-analysis, with a pooled global prevalence of 21% reported from 109 studies investigating contamination in soil samples from public spaces [3]. Humans can become infected after the accidental ingestion of these eggs. Symptoms are generally non-specific and can easily go unnoticed making it difficult to quantify the full impact of infection on the human population. Human infection with *Toxocara* is known as Toxocariasis which can manifest as distinct syndromes including ocular or visceral larva migrans [4]. To definitively diagnose a human *Toxocara* infection, biopsy material is required to isolate the larvae, which are then identified using morphological and/or molecular methods. However, performing a biopsy does not guarantee that the larvae will be present in the sample, and so this invasive method is rarely employed, leaving serological methods of detection as the primary method of diagnosis [5]. Based on serological data, another recent meta-analysis has demonstrated that up to 19% of the world's population has been exposed to *Toxocara*. Despite the large number of people infected throughout the world, our knowledge of the relationship between seropositivity and toxocariasis symptoms remains limited [6]. However, some authors have demonstrated significant associations between asthma, epilepsy, cognitive decline and *Toxocara* seropositivity [7,8]. This degree of seropositivity identifies *Toxocara* as one of the most widespread parasitic infections in the world [9]. Considering the ubiquity of this parasite in both the developed and developing world [10], it is surprising that there remains a variety of important gaps in our knowledge about the epidemiology of this disease [6,11].

One such knowledge gap is the relative importance of the different routes of transmission. Contact with soil contaminated with embryonated *Toxocara* eggs is considered the primary route of human infection, although, other routes of transmission are also possible. *Toxocara* eggs have been found on unwashed vegetables [12], in dog hair [13] and on dog owners' shoes [14], representing multiple potential routes of egg ingestion. In addition, larvae present in the tissues of infected paratenic hosts also represent a route of transmission to humans through meat consumption [15]. The relative importance of each of these routes of transmission is difficult to determine, but as some time in the environment is required before the eggs become infective, the investigation of soil contamination remains important. Studies investigating levels of soil contamination are common, and overall show that public places frequented by dogs and cats are regularly contaminated with *Toxocara* eggs [3].

In their meta-analysis investigating the global levels of soil contamination, Fakhri et al., [3] emphasised the high frequency of environmental contamination with eggs of *Toxocara*

but also highlighted pieces of information that were often lacking from these kinds of studies. Firstly, the species of *Toxocara* egg is rarely reported in such studies. Knowledge of the species responsible for environmental contamination is essential as it allows control methods to be focused on the host species responsible for the majority of the contamination. Recent modelling has suggested that pet dogs are likely to be responsible for the majority of environmental contamination in urban environments in the UK, but cats and foxes still have the potential to be important sources of *Toxocara* eggs [16], with the relative importance of different sources varying locally. It is difficult to differentiate between *T. canis* and *T. cati* eggs under the microscope, but with molecular detection methods becoming more sensitive, accurate and affordable, identification of the eggs to species level should be considered essential when conducting environmental contamination studies. Fakhri et al., [3] also drew attention to the fact that there was substantial heterogeneity between the studies included in the meta-analysis, some of which is likely attributable to variations in methodology which are often insufficiently described. A move towards standardised or at least transparent and repeatable methodology would enable direct comparison between studies and support improved understanding of factors influencing *Toxocara* spp. environmental egg density.

In this study, we aimed to use freely available geographical information system (GIS) software to guide an easily repeatable, standardised sampling strategy to establish the current level of environmental contamination with *Toxocara* spp. eggs in city parks in Dublin, Ireland. We also aimed to establish the *Toxocara* species responsible for contamination using molecular methods, as well as assessing the viability of any unembryonated eggs detected. Finally, we aimed to compare different areas within parks in terms of their degree of contamination with *Toxocara* eggs.

## Materials and methods

### Validation of egg sedimentation/flotation technique

The method of recovery from soil was based on the extraction method reported by O'Lorcain [17] and modified by Santarem [18] and Keegan and Holland [19]. The amount of soil under investigation was much higher in the current study, making it necessary to validate this modified method. Additionally, a different flotation solution (sugar/salt) was tested, as it was easier to source and dispose of safely than previously used solutions.

### Preparation of egg seeded soil

To establish the rate of recovery of *Toxocara* eggs from 50g of soil, a validation of the flotation and sieving technique was first conducted. Soil, previously determined as egg-free using the method described below, was first autoclaved to remove any undetected eggs that may have been present. This soil was then passed through a sieve with apertures of either 0.5 mm or 2 mm to assess which size was more beneficial in terms of egg recovery. Replicate conical flasks were then filled with either the 0.5 mm or 2 mm sieved soil. Stocks of *T.canis* and *T.cati* eggs were kindly provided by Christina Strube and Marie-Kristin Raulf, University of Veterinary Medicine, Hanover Germany. The eggs were recovered from experimentally infected dogs and cats as described in Kliene et al., [20]. A known number of *Toxocara* eggs, of different species (*T. canis* and *T. cati*) and different developmental stages (unembryonated & embryonated) was then added directly to the soil in each conical flask. The eggs for each treatment were counted under the microscope and added to each flask by washing them from the glass slide into the flask with drops of distilled water. The contents of the conical flask were then mixed and allowed to dry overnight.

## Flotation/sedimentation procedure

The following day, 50ml of 0.2% Tween was added to each flask. The flasks were then placed on an orbital shaker for 20 minutes at 170 RPM. The contents of each flask were then separated evenly between two 50ml centrifuge tubes. A wash bottle with distilled water was then used to carefully wash the sides of the flasks into the 50 ml tubes. The tubes were then centrifuged at 2000 RPM for 5 minutes. The supernatant was discarded, and then each tube was filled to the 45ml mark with distilled water. The contents of the tube were then mixed using two or three wooden applicator sticks to avoid the introduction of bubbles to the tube. Once sufficiently mixed, the tubes were filled to the 50ml mark with water and were centrifuged again at 2000 RPM for 5 minutes. The supernatant was again discarded, and a saturated sugar/salt flotation solution (SG ~ 1.29) was added up to the 45ml mark. The contents were again mixed using wooden applicator sticks. After mixing, the tubes were filled to the 50 ml mark with the sugar/salt solution and were centrifuged a final time at 2000 RPM for 5 minutes. The lids were then removed and using a Pasteur pipette sugar/salt flotation solution (SG 1.28) was added to create a positive meniscus at the top of the tube. A coverslip (22mm x 22mm) was then placed on each tube and allowed to stand for at least 20 minutes. The coverslip was then carefully removed from the top of the tube and placed on a glass slide. A few drops of liquid from the top of the tube were then also added to fill any space under the coverslip. The slide was then read at 100X magnification. Once one coverslip was removed, drops of flotation solution were added to again create a positive meniscus, and a second coverslip for each tube was prepared in the same way and read at 100X magnification. Results are presented as % recovery, calculated as eggs recovered/eggs seeded*100.

## Sieving procedure validation

Based on these results, the 0.5 mm sieve was chosen due to the marginally superior recovery rate. To determine if the sieving process resulted in any loss of eggs, an additional validation step was conducted. *Toxocara* egg-free, autoclaved soil (200g) was added to each of six 500 ml plastic containers. Known numbers of *T. canis* or *T. cati* eggs were added to three containers each. The contents were mixed and the containers were allowed to stand overnight. The following day, the contents of each container were passed through a 0.5 mm sieve. The first 50 g of soil passed through the sieve was then placed in a conical flask. The flotation/sedimentation procedure described above was then followed.

## Determining sample locations

Of the 66 parks under the management of Dublin City Council, 12 parks were chosen in order to provide a geographical spread of park locations, both north and south of the river Liffey, in addition to including parks of similar size and facilities. Very small parks with few facilities were usually not sampled, unless there was no other suitable park in the area. For example, parks closer to the city centre had fewer playing field facilities, so smaller parks were included in these locations. Soil samples were taken from twelve parks under the management of Dublin City Council from May 2021 to July 2022. Four different location types were chosen to be sampled in each park: entrances; playgrounds; playing field side-lines; and areas where people were observed sitting on the grass. For each location within a park, a similar sampling strategy was employed. For entrances, and sitting areas, a 15 x 15 m border was generated using QGIS 3.2 (QGIS Development Team, 2022) prior to visiting each park. These borders were placed adjacent to entrances and on areas that people could be seen sitting based on google satellite images. For the playing field side-lines, a border of 5 x 20 meters was generated using QGIS. This different shape was used to capture the smaller area people would be likely to stand along

the side-lines of playing fields. Depending on the park facilities, we aimed to sample at least two entrances, two seating areas and two playing field side-lines per park, however some parks did not have all of these location types. Once the border was generated, QGIS was then used to place 10 random coordinates within each boundary (with 2 meters at least between each sample). With the coordinates predetermined, using smartphones and google earth, these locations were then visited and soil samples were taken. As the playgrounds were highly variable in terms of their size, shape and soil substrate, predetermined coordinates could not be used to sample these locations. The judgement of the sampler was used with 10 samples taken from locations in and around the playground boundaries. Areas where people, especially children, were considered likely to interact with the soil were preferentially chosen.

## Soil sampling procedure

Once the sampling location was identified, the following sampling procedure was conducted. A bulb planter was used to take five plugs of soil, to a depth of approximately 2-5 cm, providing soil samples of about 50g each. These five samples were randomly chosen by the sampler at the predetermined GPS locations and combined in a plastic resealable bag to form one composite sample of approximately 250 g. Using a garden thermometer (Biogrod, 4 in 1 tester) the soil temperature and moisture (wet or dry) were also recorded. The sampler also recorded whether there was plant or tree cover directly above the sampling location. Grassy soil alone was not considered as having plant cover, the sampled soil needed to be covered by bushes or trees. In addition, whether or not the sampler would consider sitting down on each sampling location was also recorded. This subjective decision was based mainly on the amount of grass coverage, presence of bare soil or faeces and the general appearance of the cleanliness of the ground, but also proximity to the path and entrances. The soil samples were then returned to the lab and stored in a refrigerator for no more than three weeks before being processed. In addition, any faeces detected within the boundaries was collected and nematode eggs were recovered using the method described by Nissje *et. al.*, [21]. The day prior to processing, the approximately 250g composite soil samples were emptied from their plastic bags and placed in large trays to allow the soil to dry out. Any large clumps were broken up and large stones or pebbles were removed. Overnight was usually sufficient but two days of drying was appropriate for wetter samples. Once sufficiently dry, the samples were passed through a 0.5 mm sieve and the first 50g of soil was placed in a conical flask, after which the sedimentation/flotation technique described above was then carried out.

## Egg imaging and DNA extraction

Any detected eggs were photographed and measured using Capture 2.0 software. Once recorded and photographed, an attempt was then made to recover the egg(s). The coverslip was removed, and the egg was again located and removed using a pipette. This was a technically difficult process and occasionally resulted in the loss of the egg. The recovered eggs were then pipetted individually into microcentrifuge tubes containing 70% ethanol and stored at -20 °C.

In preparation for DNA extraction, individual eggs were transferred to microscope slides in 10 µl droplets and rehydrated in 100 µl of nuclease free water. The rehydrated eggs were then transferred in 1-2 µl droplets to 100 µl of 1x nemabiome lysis buffer. The egg was then transferred along with 10 µl of lysis buffer into a 0.2 ml tube and kept on ice. The eggs were then freeze cracked by placing them in a - 80 freezer for 10 minutes after which they were removed and allowed to defrost. This process was repeated 3 more times. Proteinase K (0.8 mg/ml) was then added to the freeze cracked individual eggs and incubated overnight at 56 °C in a water

bath followed by 15 minutes at 95 °C inactivation step. The lysates were then stored at -20 °C until the PCR was conducted.

## PCR and Sanger sequencing

PCR was performed using GoTaq G2 Hot Start polymerase (Promega, Cat: M7401) as per the manufacturer's instructions, using previously designed ascarid-specific XZ1 (5′-ATT GCGCCATCG GGTTCATTCC-3′) forward and NC2 (5′-TTAGTTTCTTTTCCTCCGCT-3′) reverse primers [22–24]. A *T. canis* specific PCR was also conducted using the previously described YY1 primer (5′-CGGTGAGCTATGCTGGTGTG-3′) and NC2 reverse and a *T. cati* specific PCR was also conducted using the previously described forward primer (5′-GGAGAAGTAAGATCGTGGCACGCGT-3′) and NC2 reverse primer. Crude DNA lysate (1 μl) was used in each 25 μl reaction. A touchdown PCR protocol as described by French et al., [25] was used for all primer sets under the following conditions: initial activation of 15 min at 95 °C, 12 cycles of 95 °C for 30 s, annealing for 30 s (starting at 60 °C, reducing by 0.5 °C per cycle), and 72 °C for 30 s, followed by a further 35 cycles of 95 °C for 30 s, 54 °C for 30 s, and 72 °C for 30 s, and final elongation at 72 °C for 7 min. PCR products were assessed by 1.3% agarose gel electrophoresis, with single bands purified by Wizard SV Gel and PCR Clean-Up System (Promega, Cat: A9281), and assessed by Nan-o Drop ND-1000 (Thermo-Fisher). Purified PCR products were sequenced using Eurofins TubeSeq service.

## Statistical analysis

Statistical analysis was conducted using Minitab (Minitab version 19, State College, USA). The numbers of *Toxocara* spp. eggs detected were compared between the twelve parks using a Kruskal-Wallis test as the data were not normally distributed. The numbers of *Toxocara* spp. eggs at each of the different locations within parks (Entrance, Playground, Playing field, Seating area) were also compared using a Kruskal-Wallis test. The Kruskal-Wallis test was performed using Minitab, which automatically adjusts the test statistic for ties using a correction factor to maintain the validity of the chi-square approximation.. Standardised absolute mean rank differences were performed to identify significant differences with pairwise comparisons following the Kruskal-Wallis Tests. The Mann-Whitney-U test was used to compare the level of contamination observed in samples that were determined to be a) wet or dry; b) with or without plant cover; c) a place where the sampler would or would not choose to sit; and d) close to visible faeces or not.

# Results

## Validation of egg sedimentation/flotation technique

The flotation/sedimentation procedure had an overall recovery rate of 49% (95% C.I. ± 4.58) when both species, embryonation stages and sieve sizes were taken into account (Table 1). Soil passed through the 0.5 mm sieve had a marginally better recovery rate than soil from the 2 mm sieve, but also resulted in coverslips that were easier to work with, as floating debris between 0.5 and 2mm occasionally made the coverslip difficult to place and read. For these reasons, the 0.5 mm sieve was chosen for further study.

## Validation of sieving procedure

To test whether the sieving process resulted in a loss of eggs, 200 g of soil was seeded with an average of 250 eggs after which the mixture was sieved and the first 50 g tested using the sedimentation/flotation technique described above. The overall recovery rate from soil that

**Table 1. Recovery rate of *Toxocara* spp. using our sedimentation/flotation technique.**

| Egg species | Stage of Development | Sieve size | Replicates | Mean Number of eggs added (range) | Mean Number of eggs Recovered (%) | Recovered % ± 95% C.I. |
|---|---|---|---|---|---|---|
| *Toxocara canis* | Unembryonated | 0.5 | 6 | 50 (46-58) | 25 (50%) | 11.3 |
| *Toxocara cati* | Unembryonated | 0.5 | 6 | 70 (43-94) | 34 (47%) | 12.9 |
| *Toxocara canis* | Embryonated | 0.5 | 3 | 63 (51-79) | 34 (55%) | 2.6 |
| *Toxocara cati* | Embryonated | 0.5 | 3 | 77 (64-100) | 41 (54%) | 7.6 |
| *Toxocara canis* | Unembryonated | 2 | 6 | 81 (60-106) | 35 (42%) | 9.0 |
| *Toxocara cati* | Unembryonated | 2 | 6 | 70 (46-103) | 37 (53%) | 10.1 |
| *Toxocara canis* | Embryonated | 2 | 6 | 58 (49-65) | 29 (48%) | 17.9 |
| *Toxocara cati* | Embryonated | 2 | 6 | 74 (55-97) | 36 (47%) | 12.1 |
| Overall mean recovery | | | | 67.5 | 33 (49%) | 4.6 |

had been seeded with known numbers of eggs prior to sieving was found to be 155% (Table 2). These results indicate that the eggs pass through the sieve preferentially in the first aliquot, resulting in a degree of egg concentration in the first 50g of soil taken from the composite sample.

A summary of the overall level of *Toxocara* spp. egg contamination of soil samples taken from Dublin City parks is provided in Table 3, with the information regarding each individual park provided in S1 Table. Overall, 4.66% of samples were found to have *Toxocara* eggs. The highest number of eggs detected in one 50g soil sample was 15, with the majority of positive samples containing one or two eggs (S1 Table). There was a significant difference in the level of contamination between the parks (H-value = 40.93, df =11, p ≤ 0.001), with two parks being more highly contaminated than the others (Fig 1). Park entrances were the most heavily contaminated locations surveyed, followed by playgrounds (Fig 2). Park entrances had significantly more *Toxocara* eggs than both playing field sidelines and areas where people were seen sitting (H-value = 25.79, df =3 p ≤ 0.001).

Additional variables that may have affected egg abundance in soil were also investigated. More eggs were recovered from locations with plant cover and those at which the researcher would not choose to sit, but no difference in egg density was observed in wet versus dry soil samples, or those taken near visible faeces (Table 4). If a faecal sample was detected within the boundaries of the soil sampling location, such samples were collected and tested for the

**Table 2. Recovery rate of Toxocara eggs following sieving of soil seeded with known numbers of *Toxocara* spp. eggs. Percent recovery is based on the fraction expected in 50/200g of soil, i.e., 25% of eggs added.**

| | Unembryonated eggs added in 200g of soil | Embryonated eggs added in 200g of soil | Total eggs in 200g of soil | Estimated eggs per 50g of soil | Unembryonated eggs recovered in first 50g(%) | Embryonated eggs recovered in first 50g (%) | Total eggs recovered in first 50g (%) |
|---|---|---|---|---|---|---|---|
| *T. canis* | 111 | 129 | 240 | 60 | 33 (119%) | 34 (105%) | 67 (112%) |
| *T. canis* | 106 | 105 | 211 | 52.75 | 39 (147%) | 36 (137%) | 75 (142%) |
| *T. canis* | 100 | 108 | 208 | 52 | 36 (144%) | 51 (189%) | 87 (167%) |
| *T. cati* | 114 | 114 | 228 | 57 | 49 (172%) | 47 (165%) | 96 (168%) |
| *T. cati* | 105 | 287 | 392 | 98 | 50 (190%) | 184 (187%) | 184 (188%) |
| *T. cati* | 100 | 142 | 242 | 60.5 | 41 (164%) | 52 (146%) | 93 (154%) |

Toxocara spp. contamination in Dublin City parks.

**Table 3. Overall summary of the level of *Toxocara* spp. egg contamination of soil samples taken from Dublin City parks. EPG = eggs per gram of soil.**

| Details | Number |
|---|---|
| Number of parks sampled | 12 |
| Number of soil samples taken | 836 |
| Number of positive samples | 39 |
| Overall % positive samples | 4.66% |
| Range of % positive samples between parks | 0 – 23% |
| Number of eggs detected | 94 |
| Number of motile larvae detected | 87 |
| % Eggs with motile larvae | 92.55% |
| Mean number of eggs per sample | 0.112 (± SE 0.0266) |
| Mean number of eggs per gram (EPG) of soil | 0.0022 (± SE 0.0005) |
| Mean number of eggs per positive sample | 2.41 (± SE 0.434) |
| Median number of eggs per positive sample | 1 |
| Mean EPG per positive sample | 0.0482 (± SE 0.009) |

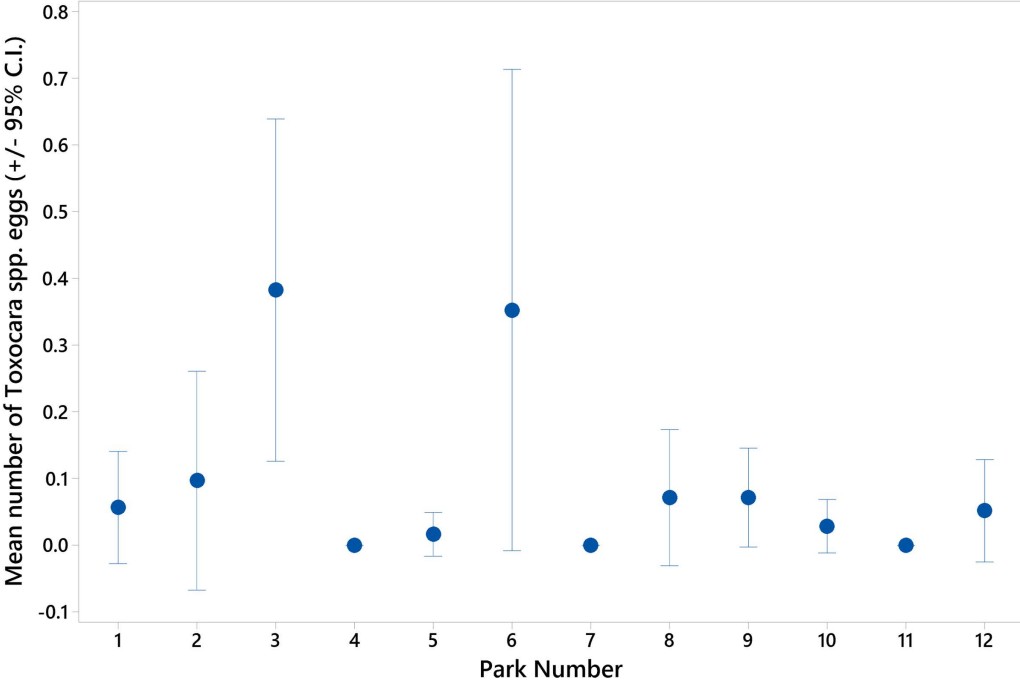

**Fig 1. Mean (± 95% Confidence Interval (C.I.) number of Toxocara eggs recovered per 50g soil sample per park.**

presence of *Toxocara* eggs. Forty-nine faecal samples were detected over the course of the study with only one sample positive for *Toxocara* giving a prevalence of 2%.

## Microscopic and molecular analysis of Toxocara spp. eggs

Of the 94 eggs detected, 55 were successfully recovered to determine the species by polymerase chain reaction (PCR). The PCR was successful for 39 individual eggs. The majority of the eggs were identified as *T. canis* (n=36), while only three *T. cati* eggs were identified. The mean size

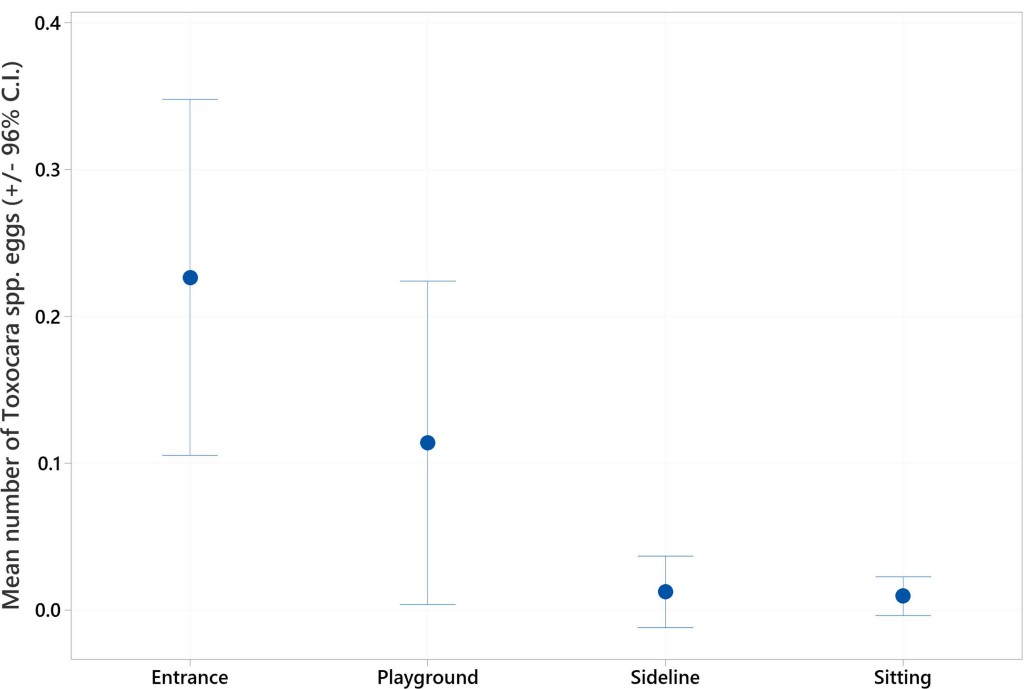

**Fig 2. Mean (± 95% C.I.) number of Toxocara eggs detected in each park location.**

**Table 4. Mann-Whitney-U tests comparing the effects of sample characteristics on the number of *Toxocara* eggs in soil.**

|  | Yes<br>*n*<br>(Mean ± S.E) | No<br>*n*<br>(Mean ± S.E) | W value | *p*-value |
|---|---|---|---|---|
| Dry Sample | 350<br>(0.101 ± 0.029) | 486<br>(0.129 ± 0.049) | 202702.5 | 0.585 |
| Would sit? | 533<br>(0.053 ± 0.022) | 303<br>(0.22 ± 0.062) | 217272.0 | <0.001 |
| Faeces visible? | 107<br>(0.047 ± 0.024) | 729<br>(0.122 ± 0.030) | 44345.0 | 0.611 |
| Plant cover? | 308<br>(0.162 ± 0.057) | 525<br>(0.083 ± 0.026) | 131659.0 | 0.025 |

Total number of samples in each category is given followed by the mean number of eggs recovered from per 50g sample.

of the 36 eggs confirmed to be *T. canis* was 89 x 76 μm (range = 73 - 100 μm length, 62 - 95 μm width). For the three *T. cati* eggs, the mean size was 78 x 63 μm (range = 67 - 84 μm length, 56 - 71 μm width). Photographs of each egg, the egg measurements and the result of the PCRs are provided in the supporting information.

## Discussion

Contamination of soil with *Toxocara* spp. eggs is an important step in parasite transmission and zoonotic disease risk. While many studies have sampled soil from public parks for eggs [3,26], few have compared the distribution of eggs from different areas within parks [27–29].

The current study sought to describe *Toxocara* egg distribution between and within parks in Dublin city, in order to inform potential control strategies. Further, the study presents a validated methodology for egg recovery from soil and a transparent method for spatial sampling of eggs.

Park entrances were the locations most heavily contaminated by *Toxocara* eggs in the parks of Dublin City. Many owners have anecdotally reported that their dogs often defecate as soon as they reach the park, and the findings of this study would support those statements, with the vast majority of eggs being found around park entrances. Other authors have also detected higher levels of dog fouling around park entrances and near car parks attached to parks [26,29]. This suggests that park entrances could be an important place to target with specific dog waste bins and signage. Interestingly, the second most contaminated location identified was playgrounds, all but one of which included in this study was surrounded by a fence. Most playgrounds also had signage indicating that dogs were not allowed in the playground areas. Molecular characterisation of the eggs found within the playgrounds indicated that dogs were most likely responsible for this contamination (noting that foxes are also sources of *T. canis* [30]). If this characterisation was not conducted, it may be assumed that cats were more likely to be responsible for this contamination due to their ability to pass through or over the fences more easily than dogs or foxes. It has also been suggested that eggs could be carried into fenced off areas as a result drainage patterns following rain. However, during sample collection, the authors of this study witnessed dog owners with puppies in fenced off playgrounds, (with signs saying no dogs allowed) on more than one occasion, suggesting puppies as likely contamination culprits. That fenced playgrounds were the second most contaminated locations, also suggests that simple signposting prohibiting dogs and dog fouling does not prevent environmental contamination. As such, future interventions need to explore other methods that encourage consistent engagement from park users.

It is striking that despite a long history of efforts to reduce dog fouling, the evidence base for their effectiveness is weak [31]. It is therefore essential that greater attempts are made to measure the impact of any anti-dog fouling intervention conducted. Despite dog-fouling being one of the most complained about issues involving the public realm [32], there are surprisingly few published studies that have measured dog fouling before and after an intervention. Success has been achieved with reductions of up to 60% in dog-fouling observed following a campaign that enlisted primary school students to draw attention to the problem in public spaces by empowering them to create posters and stencils that drew attention to and discouraged dog fouling in their local area [33,34]. In another intervention designed to reduce dog fouling, the use of a poster including eyes (which also glow in the dark), a message that cleaning up after your dog is the norm and information that you can use any bin (as opposed to specific dog fouling bins only), achieved a reduction in dog fouling of 49% across a number of sites in the UK [35]. More recently, identifying dog faeces using bright pink, environmentally friendly, spray paint to highlight dog faeces, in addition to the installation of bright, glow in the dark footprints guiding people to the nearest bins has also been shown to reduce dog fouling by 40.9% in a Slovakian park [36]. The methods described in the present study could be applied to track the effects of such interventions, if sustained, on egg contamination rates and zoonotic hazard.

In terms of the other sample characteristics measured, the moisture of the sample was not found to affect the amount of eggs recovered from the soil. Moisture is an important factor for egg development and survival [19]. The moisture measurement, taken during this work, was a one-off recording heavily influenced by the weather in the days prior to sampling. As such, this measurement may not reflect the actual impact of moisture levels on the eggs over time. The presence of faeces visible within 1 metre of the sample was also not found to impact

the number of *Toxocara* eggs detected in soil. The aim if this measurement was to associate detected faeces with the level of soil contamination in the immediate vicinity of the faecal deposit. This lack of association between faecal and soil positivity is likely due to eggs persisting in the soil long after faeces has decomposed. Overall, in this study, entrances were more commonly contaminated with faeces than the other park locations, increasing the likelihood that a proportion of the dogs defecating there may have a patent infection. Other authors have also correlated the perceived level of dog fouling with the level of egg contamination [37]. Importantly, these results demonstrate that the absence of faeces does not suggest an absence of *Toxocara* contamination.

Interestingly, more eggs were detected in locations in which the sampler would not choose to sit, (based mainly on presence of grass or bare soil, proximity to a path, and general appearance of cleanliness) possibly suggesting that accidental contact with more contaminated areas is less likely due to the unappealing appearance of the particular spot. Finally, significantly more eggs were recovered from soil samples that had some plant cover. Plant cover has been identified by other authors as having a protective effect on eggs in soil, helping to prevent the loss of moisture and to shade the eggs from direct sunlight [38,39]. Knowledge of these factors could help to guide intervention strategies, for example by mowing grass short and concentrating anti-fouling efforts near park entrances and playgrounds.

## Degree of park contamination and relationship to other studies

It is clear from the literature that the contamination of public spaces with the eggs of *Toxocara* spp. is common globally [3]. As such, the finding that 75% of parks sampled in the current study were contaminated with *Toxocara* eggs is not a surprise. This result is similar to other surveys assessing *Toxocara* soil contamination in the parks of the UK and Ireland, with park contamination rates of 74 [28] and 85% [37] reported in recent years (Table 5). These findings demonstrate that public parks remain a potentially significant source of *Toxocara* infection. In terms of the number of *Toxocara* positive soil samples overall, the level of contamination in other European countries ranges between 4 and 53%, placing our finding of 4.6% prevalence near the lower end of this range. However, the level of contamination differed significantly between parks, ranging from 0 - 23.7% (S1 Table). This low level of contamination still presents a risk, with infections caused by a few or even by just one larva of *Toxocara* spp. potentially more damaging to humans than infections with larger amounts of larvae [40].

A recent report on the public use of Dublin City Parks during the Covid Pandemic suggests that the contamination levels observed are not simply a reflection of the number of people visiting the park. Six of the ten most popular parks were sampled for *Toxocara* with four of those

**Table 5. Rates of soil contamination recorded in selected studies.**

| Reference, year sampled | Location | Positive parks (%) | Positive soil samples (%) |
|---|---|---|---|
| Current Study, 2021-2022 | Dublin, Ireland | 9/12 (75%) | 38/826 (4.6) |
| [29], 2020 | UK and Ireland | 123/142 (86%) | n/a* |
| [28], 2018 | Midlands, UK | 17/23 (74%) | 32/405 (7.9) |
| [3], various | Europe region | n/a | 1815/11834 (15) |
| [42], 2015-2016 | New York, USA | n/a | 35/91 (38.5) |
| [40], 2015 | Lisbon, Portugal | 6/12 (50) | 80/151 (53) |
| [43], 1994- 2013 | Poland | n/a | 493/3309 (14.9) |

*Sampling methodology precludes estimation.

6 parks found to be positive for *Toxocara*. Of the two parks that were heavily contaminated to a similar degree, only one features on the list of most visited parks [41], suggesting there are factors other than the number of visitors that are important in terms of environmental contamination.

### Sources of environmental contamination with Toxocara *eggs*

With the majority of eggs identified as *T. canis* by molecular methods, it is likely that dogs are responsible for the majority of the contamination in the public parks in Dublin. These results are in keeping with the modelled predictions that owned dogs likely represent the main source of environmental contamination in urban settings, due primarily to their greater densities when compared to cats and foxes, and greater associated volumes of faeces [16]. In contrast, other authors have demonstrated that *T. cati* is more common in certain localities [40,42,43]. We detected three *T. cati* eggs, in two samples, from two different park entrances. Various factors can impact which host species is responsible for the observed contamination and addressing contamination by different hosts may require some differing control strategies. As such, the molecular identification of recovered eggs should be considered an essential step in future environmental contamination studies.

Of the faecal samples recovered, only 1/49 was positive for *Toxocara* eggs. This prevalence of 2% is lower than those reported for dogs in Ireland (6% [44]) and Europe (10.8%, [1]). However, as the faecal samples in this study were found on the ground, and not taken directly from a dog, it is possible that the same dogs could have been sampled more than once, downwardly biasing prevalence. It was considered unlikely that foxes or cats were responsible for the faecal samples tested, due to the appearance and location of the deposits, further suggesting dogs as the primary cause of the contamination observed in these locations. It has also recently been reported that dog owners in Ireland often dispose of their dog's faeces inappropriately and worm their dogs insufficiently frequently to reduce the level of environmental contamination with *Toxocara* spp. [45].

### Contamination in Dublin parks over time

In terms of the degree of contamination in Dublin over time, the prevalence of positive soil samples was relatively low in the current study, with only 4.6% of samples having at least one *Toxocara* egg. The previous surveys of public parks and private gardens in Dublin carried out in 1991 and 1994 found a pooled prevalence of 13% suggesting that the level of environmental contamination in Dublin has decreased in this time [3,46]. Since those initial studies were conducted, there has been a significant reduction in the numbers of stray/unwanted dogs in Ireland. In 2002, over 21,000 dogs were euthanized in dog pounds mainly due to being unwanted, with similar numbers euthanized in the preceding years [47]. A national stray dog forum was established to address this issue and the number of stray dogs in Ireland has dramatically decreased since then, with 168 dogs euthanised in 2021 [48]. It has been difficult to estimate the dog population in Ireland, but the evidence suggests that it has at least been relatively stable since 2005 with approximately 200,000 dog licences issued in most years since then. However, dog ownership is likely to be much higher with the dog food industry estimating the dog population at over 400,000 [49]. This reduction in the number of stray dogs is likely to have had an impact on the level of environmental contamination with *Toxocara canis* eggs, and the results of the current study support this hypothesis. However, as mentioned earlier, it can be difficult to reliably compare studies using different methodologies.

Based on the experience of the authors of this study, there are a number of important methodological steps that are often poorly reported in environmental contamination studies

[3]. Specific information detailing how researchers chose their soil sample locations is often lacking, making repeating the study, or even resampling the same location to a similar degree very difficult. Once the location is chosen by the researcher, the soil sample must then be taken, which again invites considerable differences in approach. Many authors discard the top, grassy layer, which presumably may also be contaminated with *Toxocara* eggs. Some scrape an entire 1m$^2$ area while others adopt different approaches, or do not clearly state how the soil samples were obtained. Once the soil samples are acquired, the eggs must then be concentrated, and again here we see a variety of different methodologies published, with few authors also providing an established recovery rate for their specified methods. Once *Toxocara* eggs are identified from the soil samples, information regarding their developmental stage, and their viability is also of key importance, but is also not often assessed or reported. These issues have also been reported by other authors [20,50,51], and this study represents an attempt to address these reproducibility and comparability concerns.

## Proportion of eggs with motile larvae

In the present study, 93% of the recovered eggs contained motile larvae, indicating that they were potentially infective. Only seven eggs were recovered that were not fully embryonated, three of which developed motile larvae after incubation for at least two weeks at 25 °C. Only one egg was considered not viable, displaying no development after two weeks incubation (the remaining three developing eggs were lost prior to incubation). In their survey of New York City parks, Tyungu, et al., [42] reported that one locality had similarly high levels of embryonated eggs, with 70% of the eggs recovered from Bronx playgrounds found to be embryonated. However, in the same study, all eggs recovered from the other sampling locations were unembryonated. Otero et al., [40] found that 11% of the eggs they recovered from public parks in Lisbon were embryonated, and following incubation of unembryonated eggs, demonstrated that 53.1% of the detected eggs were viable, so could be considered potentially infective. The high levels of larvation in the current study suggest that conditions in Dublin parks are generally suitable for larval development and emphasise the zoonotic risk from contact with these eggs.

## Methodology and recovery rates

The results of the soil recovery method used fall within the range of values presented by other authors [20]. As mentioned earlier and by other authors, making comparisons between *Toxocara* environmental contamination studies is particularly challenging. One of the primary aims of this study was to describe a method of collection that could be repeated by others, and that would represent the same degree of sampling effort in the different locations. This was achieved for entrances and seating areas, with a similarly sized, computer-generated boundary being applied, and the same number of random sample locations assigned by the software to GPS locations that were then visited by the researchers. Using freely available GIS software to assign sampling locations, is a strategy that could be employed by other authors to ensure the same degree of sampling effort is observed between studies. We also used a bulb planter, a tool that can be easily carried and used to dig out a standardised amount of soil required for egg recovery in an easily repeatable manner. Additionally, we did not remove the top layer of grass/soil as many other authors report, but instead excluded these components following drying and sieving. The overall recovery rate of 49% established in the method validation suggests that the number of eggs actually present in samples in the present study was approximately twice that estimated. Notably, egg recovery was measured using seeded samples, and was not affected by the species or embryonation status of the eggs. An additional validation

step was also employed as it was hypothesised that the soil/sieve combination could result in the loss of eggs, with their sticky nature leaving them attached to the debris remaining in the sieve. Contrary to our expectations, the eggs were not evenly distributed among the sieved aliquots but demonstrated a degree of concentration in the first 50g of sieved soil. This suggests that the recovery rate could be greater than 49% when this dry sieving step is included. This unexpected finding demonstrates the value of carrying out a validation of employed recovery techniques.

## Conclusion

In Ireland and around the world, the widespread contamination of public places with the eggs of *Toxocara* spp. represents a potential risk to public health. Interventions designed to reduce dog fouling and prevent egg excretion through regular deworming of pet dogs and cats are required. With the majority of contamination detected around park entrances, these locations could be selected to concentrate dog fouling reduction interventions. However, these interventions need to be coupled with studies measuring the level of dog fouling and environmental contamination with *Toxocara* eggs, to ensure that they are having the desired effect. Here we report a standardised sampling strategy using freely available GIS software to allow researchers to conduct environmental sampling in different locations, to a similar degree of intensity. Using a strategy like this should allow for more valid comparisons between studies conducted at different locations and at different times.

## Supporting information

**S1 Table. Summary of the results for each park.**
(DOCX)

**S1 Data. Excel spreadsheet containing all the data related to soil sampling (sheet 1, Full Dataset) as well as all of the egg related information (sheet 2, Egg information).**
(XLSX)

**S1 File. Egg photos and measurements.** Photographs and measurements of the eggs recovered as part of the study.
(ZIP)

## Acknowledgments

The authors would like to thank Christina Strube and Marie-Kristin Raulf for providing the *Toxocara* eggs to conduct the recovery method validation. We thank Dublin City Council for their permission to conduct this research in their beautiful parks. We would also like to thank all the interested park goers for chatting to us while collecting our soil samples. Finaly, we would like to thank the Irish Research Council for funding this research.

## Author contributions

**Conceptualization:** Jason Keegan, Paul M. Airs, Eric R Morgan, Celia V. Holland.

**Data curation:** Jason Keegan, Anya Rishi Dingley.

**Formal analysis:** Jason Keegan.

**Funding acquisition:** Jason Keegan, Celia V. Holland.

**Investigation:** Jason Keegan, Anya Rishi Dingley, Conor Courtney, Celia V. Holland.

**Methodology:** Jason Keegan, Paul M. Airs, Claire Brown, Anya Rishi Dingley, Conor Courtney, Eric R Morgan, Celia V. Holland.

**Project administration:** Jason Keegan, Celia V. Holland.

**Supervision:** Jason Keegan, Celia V. Holland.

**Visualization:** Jason Keegan.

**Writing – original draft:** Jason Keegan.

**Writing – review & editing:** Jason Keegan, Paul M. Airs, Claire Brown, Anya Rishi Dingley, Conor Courtney, Eric R Morgan, Celia V. Holland.

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
