## [Decision Letter · Decision Letter 0]

12 Dec 2024

PNTD-D-24-01372Park entrances, commonly contaminated with infective Toxocara canis eggs, present a risk of zoonotic infection and an opportunity for focused intervention.PLOS Neglected Tropical Diseases Dear Dr. Keegan, Thank you for submitting your manuscript to PLOS Neglected Tropical Diseases. After careful consideration, we feel that it has merit but does not fully meet PLOS Neglected Tropical Diseases's publication criteria as it currently stands. Therefore, we invite you to submit a revised version of the manuscript that addresses the points raised during the review process. Please submit your revised manuscript within 30 days Feb 10 2025 11:59PM. If you will need more time than this to complete your revisions, please reply to this message or contact the journal office at plosntds@plos.org. Please include the following items when submitting your revised manuscript: * A rebuttal letter that responds to each point raised by the editor and reviewer(s). You should upload this letter as a separate file labeled 'Response to Reviewers '. This file does not need to include responses to any formatting updates and technical items listed in the 'Journal Requirements' section below. * A marked-up copy of your manuscript that highlights changes made to the original version. You should upload this as a separate file labeled 'Revised Manuscript with Track Changes '. * An unmarked version of your revised paper without tracked changes. You should upload this as a separate file labeled 'Manuscript '. If you would like to make changes to your financial disclosure, competing interests statement, or data availability statement, please make these updates within the submission form at the time of resubmission. Guidelines for resubmitting your figure files are available below the reviewer comments at the end of this letter. We look forward to receiving your revised manuscript. Kind regards, Amy J Davis, Ph.D.Academic EditorPLOS Neglected Tropical Diseases Krystyna CwiklinskiSection EditorPLOS Neglected Tropical Diseases

Shaden Kamhawi

co-Editor-in-Chief

Paul Brindley

co-Editor-in-Chief

**Additional Editor Comments :** Both reviewers and I agree this paper would make a nice contribution to the literature. The manuscript is well written and well-reasoned. There are a few comments that the authors need to address.

How were the twelve parks selected? How many parks were possible to be sampled from? In the statistical methods, please be clearer about what is meant by the ‘level of contamination’. What exactly is the response variable here? Ensure that the wording is consistent throughout from methods, results, and tables/figures. How were Kruskal-Wallis tests adjusted for ties and why? In the conclusion paragraph, be clear in the first sentence that you are referring to the general widespread contamination and not that this is a result from this study. **Journal Requirements:**

At this stage, the following Authors/Authors require contributions: Jason Keegan, Paul Airs, Claire Brown, Anya Rishi Dingley, Conor Courtney, Eric R Morgan, and Celia Holland. Please ensure that the full contributions of each author are acknowledged in the "Add/Edit/Remove Authors" section of our submission form.

- ® on page: 10.

5) We have noticed that there is a reference to supplementary table 5 on pages 20 and 21. However, there is no corresponding file uploaded to the submission. Please upload it as a separate file with the item type 'Supporting Information', or if it is no longer to be included, please remove any references to it within the manuscript.

6) We have noticed that you have uploaded Supporting Information files, but you have not included a list of legends. Please add a full list of legends for your Supporting Information files after the references list.

7) We note that your Data Availability Statement is currently as follows: "All relevant data are within the manuscript and its Supporting Information files.". Please confirm at this time whether or not your submission contains all raw data required to replicate the results of your study. Authors must share the “minimal data set” for their submission. PLOS defines the minimal data set to consist of the data required to replicate all study findings reported in the article, as well as related metadata and methods (https://journals.plos.org/plosone/s/data-availability#loc-minimal-data-set-definition).

8) Please amend your detailed Financial Disclosure statement. This is published with the article. It must therefore be completed in full sentences and contain the exact wording you wish to be published.

Please also include the grant number in the Financial Disclosure statement.

**Reviewers' comments:** Reviewer's Responses to Questions

**Key Review Criteria Required for Acceptance?**

**Methods**

-Are the objectives of the study clearly articulated with a clear testable hypothesis stated?

-Is the study design appropriate to address the stated objectives?

-Is the population clearly described and appropriate for the hypothesis being tested?

-Is the sample size sufficient to ensure adequate power to address the hypothesis being tested?

-Were correct statistical analysis used to support conclusions?

-Are there concerns about ethical or regulatory requirements being met?

Reviewer #1: YES

Reviewer #2: no concerns, excellent author included actual egg sizes

**Results**

-Does the analysis presented match the analysis plan?

-Are the results clearly and completely presented?

-Are the figures (Tables, Images) of sufficient quality for clarity?

Reviewer #1: YES

Reviewer #2: Table 2 questions: in final 3 columns data was presented as %’s but all were entered as >100% I was confused if this was intentional

**Conclusions**

-Are the conclusions supported by the data presented?

-Are the limitations of analysis clearly described?

-Do the authors discuss how these data can be helpful to advance our understanding of the topic under study?

-Is public health relevance addressed?

Reviewer #1: YES

Reviewer #2: -Are the conclusions supported by the data presented? yes

-Are the limitations of analysis clearly described? yes

-Do the authors discuss how these data can be helpful to advance our understanding of the topic under study? yes

-Is public health relevance addressed? yes!

**Editorial and Data Presentation Modifications?**

Reviewer #1: ACCEPT

Reviewer #2: See Results

**Summary and General Comments**

Reviewer #1: This manuscript provides a useful guide for others conducting similar research on the contamination of parks and public places with dog and cat feces as a means of identifying potential areas for control. the manuscript is well written, referenced and remarkably free of errors. These types of studies have been conducted in many parts of the world usually without using molecular tools nor the rigor included in this paper. The authors are commended for their attention to detail. Such a paper should hopefully elevate similar studies conducted in other parts of the world.

Reviewer #2: (No Response)

PLOS authors have the option to publish the peer review history of their article (what does this mean? ). If published, this will include your full peer review and any attached files.

**Do you want your identity to be public for this peer review?** For information about this choice, including consent withdrawal, please see our Privacy Policy .

Reviewer #1: No

Reviewer #2: **Yes: ** Donna Lynn Tyungu

---

## [Editor Report · Decision Letter 1]

12 Feb 2025

Dear Dr Keegan,

We are pleased to inform you that your manuscript 'Park entrances, commonly contaminated with infective Toxocara canis eggs, present a risk of zoonotic infection and an opportunity for focused intervention.' has been provisionally accepted for publication in PLOS Neglected Tropical Diseases.

Best regards,

Amy J Davis, Ph.D.

Academic Editor

Krystyna Cwiklinski

Section Editor

Shaden Kamhawi

co-Editor-in-Chief

Paul Brindley

co-Editor-in-Chief

The authors have done a nice job responding to reviewer comments.

---

## [Editor Report · Acceptance letter]

Dear Dr Keegan,

We are delighted to inform you that your manuscript, "Park entrances, commonly contaminated with infective Toxocara canis eggs, present a risk of zoonotic infection and an opportunity for focused intervention.," has been formally accepted for publication in PLOS Neglected Tropical Diseases.

Best regards,

Shaden Kamhawi

co-Editor-in-Chief

Paul Brindley

co-Editor-in-Chief
